# Progressive Learning for Physics-informed Neural Motion Planning

Ruiqi Ni and Ahmed H. Qureshi
Department of Computer Science, Purdue University
{ni117,ahqureshi}@purdue.edu

Start · Intermediate · Goal

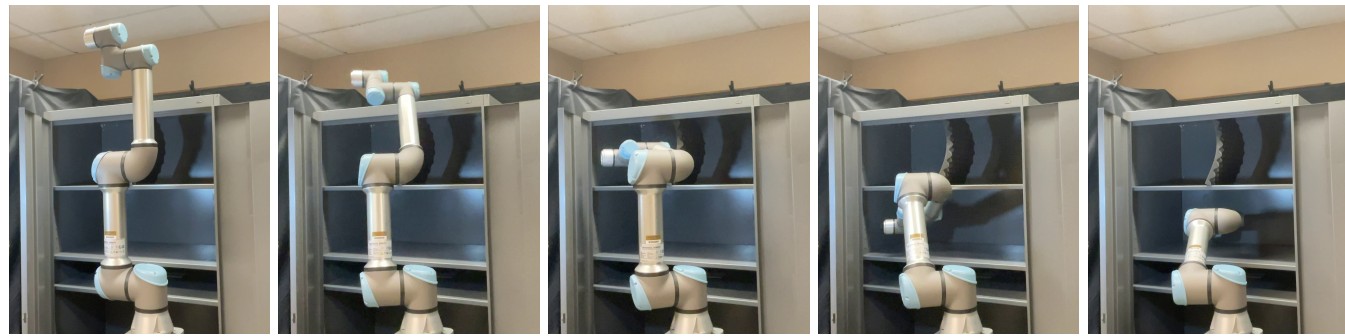

Fig. 1: Physics-informed neural motion planning of a 6-DOF robot manipulator in a real-world narrow passage environment. The images from left to right show the robot's motion sequence from its start to the desired goal configuration. In this case, the proposed approach took 0.05 seconds, whereas LazyPRM* took 2.79 seconds to find a path, making our method at least 50× faster than a traditional approach.

*Abstract*—**Neural motion planners (NMPs) demonstrate fast computational speed in finding path solutions but require a huge amount of expert trajectories for learning, thus adding a significant training computational load. In contrast, recent advancements have also led to a physics-informed NMP approach that directly solves the Eikonal equation for motion planning and does not require expert demonstrations for learning. However, experiments show that the physics-informed NMP approach performs poorly in complex environments and lacks scalability in high-dimensional real robot settings. To overcome these limitations, this paper presents a novel and tractable Eikonal equation formulation and introduces a new progressive learning strategy to train neural networks without expert data in complex, cluttered, high-dimensional robot motion planning scenarios. We show that our approach scales to the real robot set up in a narrow passage environment. The proposed method's videos and code implementations are available at https://github.com/ruiqini/P-NTFields.**

## I. INTRODUCTION

Robots moving in their surrounding environment must find their feasible motion trajectory coordinating their actuators to move from their start configuration to goal configuration while satisfying all the constraints, such as collision avoidance. Various approaches exist, from classical methods [16, 19, 7, 9, 3, 6] to learning-based neural motion planners (NMPs) [12, 13, 5, 11, 8, 1], that solve motion planning problems.

Inspired by physics-informed deep learning models [15, 17] and Fast Marching Method (FMM) [16, 19] for motion planning, recent development has led to a physics-informed NMP called Neural Time Fields (NTFields) [10] that require no expert training trajectories and instead directly learn to

solve the Eikonal equation for motion planning. Once trained, NTFields output the speed and time fields in the given environment for the desired start and goal configuration. Time fields' gradients are then followed to retrieve the feasible path solution for the underlying MP problem. Although NTFields find path solutions extremely fast and require no expert data, they struggle in complex environments and do not scale well to high-dimensional planning problems. These limitations are mainly due to the following two reasons. First, the Eikonal equation formulation has an extremely sharp feature solution around low-speed obstacles, making it difficult for the underlying deep-learning model to converge and perform well in complex scenarios. Second, training deep neural models to solve PDEs is inherently challenging and requires advanced learning strategies and an expressive PDE formulation with a smooth loss landscape.

Therefore, this paper addresses the limitations of NTFields and proposes a new progressive learning method, which also requires no training trajectories and scales very well to complex scenarios, including high-dimensional, real-world robot manipulator planning problems. The main contributions of the paper are summarized as follows:

- We highlight that the Eikonal equation formulation for motion planning in NTFields can converge to incorrect local minimums during training, resulting in relatively low performance and incapability to scale to complex environments.
- We introduce a novel progressive speed scheduling strategy that iteratively guides neural model training from a

constant high speed to a very low speed around obstacles in the environment, preventing incorrect local minimums when training physics-informed NMPs in complex, cluttered environments.

- We propose using the viscosity term [2] based on the Laplacian operator in the Eikonal equation formulation to transform its ill-posed, non-linear behavior into a semi-linear elliptic representation with a unique smooth solution around low-speed obstacles. Our novel formulation leads to physics-informed NMPs that are scalable to complex scenarios.

- We also demonstrate our framework performance using a 6 degree-of-freedom (DOF) UR5e robot in solving real-world narrow passage motion planning problems, as shown in Fig. 1.

## II. BACKGROUND

This section formally presents the background to robot motion planning problems and their solutions through physics-informed NMPs.

### A. Robot Motion Planning

Let the robot's configuration and environment space be denoted as $\mathcal{Q} \subset \mathbb{R}^d$ and $\mathcal{X} \subset \mathbb{R}^m$, where $\{m, d\} \in \mathbb{N}$ represents their dimensionality. The obstacles in the environment, denoted as $\mathcal{X}_{obs} \subset \mathcal{X}$, form a formidable robot configuration space (c-space) defined as $\mathcal{Q}_{obs} \subset \mathcal{Q}$. Finally, the feasible space in the environment and c-space is represented as $\mathcal{X}_{free} = \mathcal{X} \backslash \mathcal{X}_{obs}$ and $\mathcal{Q}_{free} = \mathcal{Q} \backslash \mathcal{Q}_{obs}$, respectively. The objective of robot motion planning algorithms is to find a trajectory $\tau \subset \mathcal{Q}_{free}$ that connects the given robot start $q_s \in Q_{free}$ and goal $q_g \in Q_{free}$ configurations. Furthermore, additional constraints are sometimes imposed on the trajectory connecting the start and goal, such as having the shortest Euclidean distance or minimum travel time. The latter is often preferred as it allows imposing speed constraints near obstacles for robot and environment safety. However, planning under speed constraints is computationally expensive, and existing methods rely on path-smoothing techniques when safety is desired.

### B. Physics-informed Motion Planning Framework

Recent development led to a physics-informed motion planning framework called Neural Time Fields (NTFields) [10], which provide a computationally-efficient and demonstration-free deep learning method for motion planning problems. It views motion planning problems as the solution to a PDE, specifically focusing on solving the Eikonal equation. The Eikonal equation, a first-order non-linear PDE, allows finding the shortest trajectory between start ($q_s$) and goal ($q_g$) under speed constraints by relating a predefined speed model $S(q)$ at configuration $q_g$ to the arrival time $T(q_s, q_g)$ from $q_s$ to $q_g$ as follows:

$$1/S(q_g) = \|\nabla_{q_g} T(q_s, q_g)\| \tag{1}$$

The $\nabla_{q_g} T(q_s, q_g)$ is the partial derivative of the arrival time $T(q_s, q_g)$ function with respect to $q_g$. Therefore, finding a

trajectory connecting the given start and goal requires solving the PDE under a predefined speed model and arrival time function. The arrival time function in NTFields is factorized as follows:

$$T(q_s, q_g) = \|q_s - q_g\|/\tau(q_s, q_g) \tag{2}$$

The $\tau(q_s, q_g)$ is the factorized time field which is the output of NTFields' deep neural network for the given $q_s$ and $q_g$. Since the neural network in NTfields outputs the factorized time field $\tau$, the corresponding predicted speed is computed using the above equation. Furthermore, the NTField framework determines the ground truth speed using a predefined speed function:

$$S^*(q) = \frac{s_{const}}{d_{max}} \times \text{clip}(\mathbf{d}(\mathbf{p}(q), \mathcal{X}_{obs}), d_{min}, d_{max}) \tag{3}$$

where $\mathbf{d}(\cdot, \cdot)$ is the minimal distance between robot surface points $\mathbf{p}(q)$ at configuration $q$ and the environment obstacles $\mathcal{X}_{obs}$. The $d_{min}$, and $d_{max}$ are minimum and maximum distance thresholds, and the $s_{const}$ is a predefined speed constant; we normalize $s_{const} = 1$ to represent the maximum speed in the free space, and $s_{min} = s_{const} \times d_{min}/d_{max}$ represents the minimum speed in the obstacle space. Finally, the NTFields neural framework is trained end-to-end using a isotropic loss function between predicted $S$ and ground truth $S^*$ speeds.

## III. PROPOSED METHOD

Although NTFields demonstrate the ability for efficient motion planning without expert training data, it exhibits relatively low success rates in complex, cluttered environments, including high-dimensional problems. We observed that these limitations are mainly because of the ill-posed nature of the Eikonal equation and that the physics-informed loss landscapes are hard to optimize in general. To overcome these limitations, we introduce a new progressive learning algorithm comprising a novel viscosity-based Eikonal equation formulation and a progressive speed update strategy to train physics-informed NMPs in complex, high-dimensional scenarios.

### A. Viscosity-based Eikonal Equation

The Eikonal equation's exact solution has several problems that lead to neural network fitting issues. First, the solution is not differentiable at every point in space, which means a neural network cannot approximate the solution very well, especially for the sharp feature in low-speed environments. Second, the gradient $\nabla_{q_g} T(q_s, q_g)$ is not unique at these non-smooth points, which will also cause the neural network fitting issue because training is based on the supervision of the gradient $\nabla_{q_g} T(q_s, q_g)$.

To fix these problems, we propose to use a viscosity term that can provide a differentiable and unique approximation of the Eikonal equation's solution. The viscosity term comes from the vanishing viscosity method [2]. It adds the Laplacian $\Delta_{q_g} T(q_s, q_g)$ to the Eikonal equation, i.e.,

$$1/S(q_g) = \|\nabla_{q_g} T(q_s, q_g)\| + \epsilon \Delta_{q_g} T(q_s, q_g), \tag{4}$$

where $\epsilon \in \mathbb{R}$ is a scaling coefficient. The resulting system in Eq. 4 is a semi-linear elliptic PDE with a smooth and unique

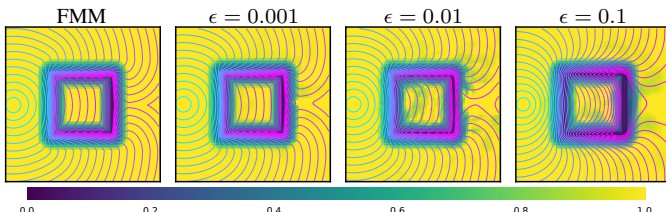

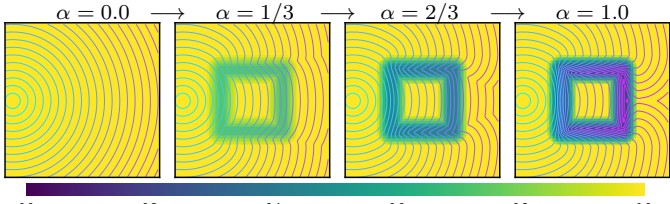

Fig. 2: Effect of viscosity coefficient, $\epsilon$, on the correctness of time field results. It can be seen a large value of $\epsilon$ deviates from the solution given by the expert. The expert is FMM which finds a solution to the Eikonal equation. The colorbar shows the speed fields range from 0 to 1.

solution. Furthermore, the value of $\epsilon$ affects the smoothness of the predicted time fields. In Fig 2, we compare fields with different values of $\epsilon$ to the ground truth field generated with the FMM approach. It can be seen that by varying the $\epsilon$, the correctness of results varies compared to the ground truth. In practice, when the coefficient $\epsilon \rightarrow 0$, the smooth and unique solution of Eq. 4 will approach the exact solution of the Eikonal equation Eq. 1.

### B. Progressive speed scheduling

This section introduces our progressive speed scheduling approach to train physics-informed motion planners in complex environments. The physics-based loss functions are generally challenging to optimize as they depend on the gradient of the underlying neural network. In physics-informed motion planners, the optimization becomes more difficult due to low-speed conditions near obstacles, often leading to an incorrect local minimum, i.e., despite small training loss, the neural model behaves as if low-speed obstacles do not exist in the environment. To circumvent the incorrect local minimums, we observe and leverage the following two properties of the Eikonal equation to progressively guide the NN training process and capture the low-speed obstacle space for collision avoidance.

First, we notice the solution of the Eikonal equation (Eq. 1), $T(q_s, q_g)$, in a constant max speed scene ($S(q) = 1$) will become the distance between the given start and goal, which leads to trivial solution $\tau(q_s, q_g) = 1$. Second, we find that the interpolation from the constant max-speed to the low speed around obstacles is continuous, and the solutions of the Eikonal equation along those interpolations are also continuous. Based on these observations, we propose a progressive speed alteration strategy that gradually scales down the speed from a constant max value to a low value around obstacles using a parameter $\alpha(t) \in [0, 1]$, i.e.,

$$S^*_{\alpha(t)}(q) = (1 - \alpha(t)) + \alpha(t)S^*(q), \tag{5}$$

where $t \in \mathbb{N}$ represent the training epochs. Therefore, when $\alpha(t) = 0$, the scene will have a constant max speed, and the Eikonal equation solution will be trivial. Furthermore, when $\alpha(t) = 1$, the scene will have low speed around obstacles. Fig 3 shows the gradual progression of speed and time fields as $\alpha$ linearly scales from 0 to 1. It can be seen that the speed

Fig. 3: Progressively decreasing the speed around obstacles using parameter $\alpha$ leads to continuous interpolation of speed and time fields in the given environment. The colorbar shows the speed fields range from 0 to 1.

and time fields are changing continuously with $\alpha$ changing linearly.

To train the physics-informed motion planner, we start with a low value of $\alpha(t)$ and let NN fit a constant speed trivial solution. Next, we progressively interpolate the field from constant max speed to low speed by gradually increasing the $\alpha(t)$ over the training epochs. The NN can easily fit the trivial solution. Then progressively decreasing obstacle speed $S^*(q)$ guides the network to learn the interpolating lower-speed fields. Furthermore, we also observe that the speed fields change linearly with $\alpha(t)$, but the resulting time fields change more aggressively. Thus, we also reduce the rate of change of $\alpha(t)$ as the training epochs increase.

### C. Neural Architecture

This section describes our neural framework, as shown in Fig. 4, for generating the speed and time fields for solving the robot motion planning problems. Our framework comprises the following modules. Given the robot's initial ($q_s$) and target ($q_g$) configurations, we use random Fourier features $\gamma$ [18, 14] for obtaining high-frequency robot configuration embeddings. These features are further processed into a latent embedding by a C-space encoder $f(\cdot)$, which is a ResNet-style multi-layer perception [4]. To combine features $f(\gamma(q_s))$ and $f(\gamma(q_g))$, we use the non-linear symmetric operator $\bigotimes$ from NTFields method [10], i.e. $f(\gamma(q_s)) \bigotimes f(\gamma(q_g)) = [\max(f(\gamma(q_s)), f(\gamma(q_g))), \min(f(\gamma(q_s)), f(\gamma(q_g)))]$.

Our time field generator network $g$ is a ResNet-style multi-layer perceptron which takes the encoding $f(\gamma(q_s)) \bigotimes f(\gamma(q_g))$ and outputs the factorized time field $\tau(q_s, q_g) = g(f(\gamma(q_s)) \bigotimes f(\gamma(q_g)))$. Given the $\tau(q_s, q_g)$, we compute its gradient and Laplacian to determine the $S(q_s)$ and $S(q_g)$. Finally, we propose a smooth isotropic objective function 6 to train our framework.

$$L(S^*_\alpha(q), S(q)) = \frac{S^*_\alpha(q_s)}{S(q_s)} + \frac{S(q_s)}{S^*_\alpha(q_s)} + \frac{S^*_\alpha(q_g)}{S(q_g)} + \frac{S(q_g)}{S^*_\alpha(q_g)} - 4 \tag{6}$$

### D. Planning pipeline

Once trained, we use the execution pipeline similar to the NTFields method. First, we predict $\tau(q_s, q_g)$ for the given start $q_s$, goal $q_g$. Next, the factorized time, $\tau$, parameterizes Eq. 2 and 1 for computing time $T(q_s, q_g)$ and speed fields $S(q_s), S(q_g)$, respectively. Finally, the path solution is determined in a bidirectional manner by iteratively updating the

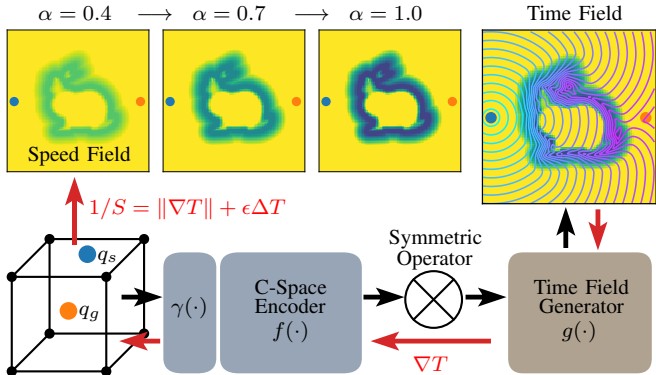

$$\alpha = 0.4 \quad\longrightarrow\quad \alpha = 0.7 \quad\longrightarrow\quad \alpha = 1.0 \qquad \text{Time Field}$$

Speed Field

$$1/S = \|\nabla T\| + \epsilon \Delta T$$

$q_s$

$q_g$

$\gamma(\cdot)$

C-Space Encoder $f(\cdot)$

Symmetric Operator

Time Field Generator $g(\cdot)$

$\nabla T$

Fig. 4: The neural architecture comprises the Fourier-based C-space Encoder, symmetric operator, and time-field generator. Three images on the top left show we progressively decrease the speed around a bunny-shaped obstacle to guide the neural network training. The image on the top right shows the final time field from start to goal generated by the trained model.

start and goal configurations as follows,

$$q_s \leftarrow q_s - \beta S^2(q_s)\nabla_{q_s} T(q_s, q_g)$$
$$q_g \leftarrow q_g - \beta S^2(q_g)\nabla_{q_g} T(q_s, q_g) \tag{7}$$

The parameter $\beta \in \mathbb{R}$ is a predefined step size. Furthermore, at each planning iteration, the start and goal configurations are updated using gradients to march toward each other until $\|q_s - q_g\| < d_g$, where $d_g \in \mathbb{R}$.

## IV. EVALUATION

In this section, we evaluate our method through the 6-DOF UR5e robot manipulator planning in two complex cabinet environments with narrow passages. For these scenarios, we present evaluations in both simulation and real-world.

In the simulation, we directly load a cabinet mesh, whereas, for real setup, we use Dot3D with RealSense camera to scan and create a point cloud of an actual cabinet. To form our test set, we randomly sampled $2\times100$ start and goal configuration pairs for simulated and real-world environments.

The table in Fig. 5 compares our method, NTField, RRT*, Lazy-PRM*, and RRT-Connect in both scenarios. We exclude IEF3D due to large data generation and training times. In the table, it can be seen that our method achieves the highest success rate with the shortest execution time, demonstrating the effectiveness of our progressive learning approach in complex, narrow passage environments.

Fig. 5 shows the execution of our method (left) and RRT-Connect (right) in a challenging case in the simulated environment and the table underneath presents the overall statistical comparison of the indicated methods on the testing dataset. In the presented scenario, the UR5e robot's end effector starts from the middle shelf of the cabinet and crosses two relatively thin obstacles to the bottom shelf of the cabinet without collision. In this particular situation, NTField could not find a solution whereas our method took 0.07 seconds to get a 0.83 length path with a safe margin of 0.03, and RRT-Connect took 20.13 seconds to get a 0.90 length path with a safe margin of 0.02. For real-world experiments, in Fig. 1, we show a

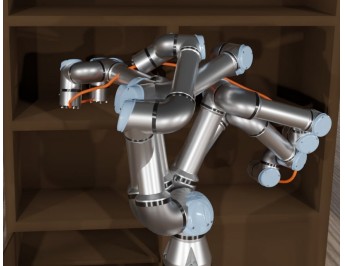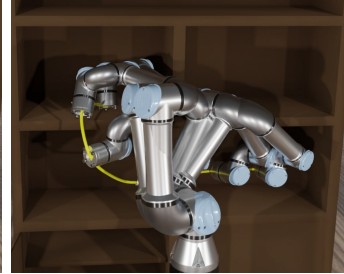

| Manipulator | time (sec) | length | safe margin | sr(%) |
|---|---|---|---|---|
| Ours | $0.03 \pm 0.00$ | $0.43 \pm 0.10$ | $0.04 \pm 0.00$ | 92.0 |
| NTFields | $0.05 \pm 0.00$ | $0.38 \pm 0.06$ | $0.04 \pm 0.00$ | 84.5 |
| RRT* | $5.16 \pm 0.01$ | $0.52 \pm 0.36$ | $0.04 \pm 0.00$ | 67.0 |
| LazyPRM* | $2.79 \pm 0.48$ | $0.76 \pm 0.80$ | $0.04 \pm 0.00$ | 86.0 |
| RRT-Connect | $1.08 \pm 0.69$ | $1.14 \pm 0.23$ | $0.02 \pm 0.00$ | 87.5 |

Fig. 5: Our method (left) and RRT-Connect (right) in a challenging case in the simulated environment: the manipulator crosses two relatively thin obstacles to move from the middle (start) to the bottom (goal) shelf. The table shows statistical results on $2\times100$ different starts and goals for two environments.

challenging path that the robot went from the initial pose to make its end effect go deep into the cabinet.

## V. DISCUSSIONS, CONCLUSIONS, AND FUTURE WORK

We propose a novel progressive learning framework to train physics-informed NMPs by solving the Eikonal equation without expert demonstration. Our method deals with the PDE-solving challenges in physics-informed NMPs such as NT-Fields [10]. First, we propose a progressive speed scheduling strategy that begins with finding a simple PDE solution at constant high speed and then gradually decreases the speed near the obstacle for finding a new solution. Second, we propose to use the viscosity term for the Eikonal equation and convert a nonlinear PDE to a semi-linear PDE, which is easy for a neural network to solve. Thus our method solves the Eikonal equation more precisely and efficiently and increases the overall performance in solving motion planning problems than prior methods. Additionally, our method requires fewer neural network parameters due to our progressive learning strategy than NTFields, leading to computationally efficient physics-informed NMPs' training and planning. Furthermore, we also demonstrate that our method scales to complex scenarios, such as real-world narrow-passage planning with a 6-DOF UR5e manipulator.

Although our method can scale to complex real-world setups and outperform prior methods with expert demonstration data, a few limitations, highlighted in the following, will still be the focus of our future research directions. First, our method cannot generalize to unseen environments. Therefore, one of our future directions will be to explore novel environment encoding strategies to make physics-informed NMP generalize to the novel, never-before-seen environments. Lastly, aside from addressing a few limitations, we also aim to explore novel PDE formulations to train physics-informed NMPs to solve motion planning under dynamic and manifold constraints.

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
