# OpenReview forum: "Progressive Learning for Physics-informed Neural Motion Planning"
_roboticsfoundation.org/RSS/2023/Workshop/Symmetry — RSS 2023 Workshop Symmetry Oral_

### Official Review · Reviewer_7cWc · 2023-06-16
**Well motivated improvements of Neural Time Fields with a few presentation issues**

**Rating:** 8
**Confidence:** 4

**Review:**

The authors propose two extensions to their previous method, NTFields [2]. One is to avoid local minima during training by going from fitting a constant speed field towards a field with low speeds around obstacles. The other is to use an additional viscosity term [1] to get unique gradients in those low-speed areas.

Strengths:
- The targeted limitations of the baseline method are clearly identified and their importance well conveyed.
- The progressive speed scheduling is nicely illustrated and I find it intuitive to follow along the explanation of the proposed solution.
- The experiments show a clear increase in success rate while maintaining approximately the same runtime and path lengths as the underlying method [2].

Weaknesses:
- At several occasions I felt the need to pull up the original paper to clarify some key parts. In my mind, even if it's a (short version of) follow-up, it should be as self-contained as possible. For example, how is the non-linear symmetric operator defined? Which parts of III.C are "ours" as in "newly proposed" and which are "ours" as in "coming from our previous work"? What does FMM (Fast Marching Method) stand for (and cite the related papers)?
- An abstract on related work (e.g., in the intro) is missing in my opinion. There are baselines that the method is compared against after all.
- I find the claim of solving "complex, cluttered, multiple high-dimensional robot motion planning scenarios" is currently not well supported by the provided experiments. The evaluation is carried out on two shelf models in simulation and for planning within one real shelf - "cluttered" involves multiple object in my view, "multiple" is more than 3 shelves to me. I would suggest to weaken the language on that part (see abstract, intro, conclusion).

I would encourage the authors to address these writing/presentation related weaknesses so that readers can focus on the nice ideas proposed herein.

---

### Official Review · Reviewer_xVpH · 2023-06-20
**Review of Progressive Learning for Physics-informed Neural Motion Planning**

**Rating:** 9
**Confidence:** 4

**Review:**

The paper introduces a new progressive learning approach to training physics-informed NMPs in complex scenarios. The proposed method addresses the limitations of NTFields by introducing a progressive learning framework that can scale to complex environments without the need for expert demonstrations. The framework incorporates a progressive speed scheduling strategy and a viscosity term to solve the Eikonal equation more accurately and efficiently. The paper showcases the framework's performance by solving real-world narrow passage motion planning problems and demonstrates that the proposed method is at least 50 times faster than traditional approaches and achieves higher success rates.

Pros:
1. The paper presents a novel progressive learning framework for training physics-informed NMPs efficiently, scalable to complex environments, and without the need for expert demonstrations. This has the potential to improve the efficiency and effectiveness of motion planning for various robotic applications.
2. Evaluation shows that the method achieves a higher success rate and improved efficiency compared to traditional approaches.

Cons:
1. The generalizability of the proposed method to other types of robots or environments beyond the narrow passage scenario is not explicitly discussed. Further experiments are needed to sufficiently demonstrate the method’s ability to handle complex, cluttered, and high-dimensional robot motion planning scenarios.
2. The specific limitations of the physics-informed NMP approach in complex environments are not explicitly mentioned in the paper, which limits a full assessment of the method's limitations.

---

### Decision · Program_Chairs · 2023-06-24

**Decision:**

Accept (Oral)

**Comment:**

Congratulations! We encourage the authors to revise the paper based on the reviewer's feedback.
Your paper will be presented as both a long oral presentation and a poster. Detailed instructions about the presentation format and camera-ready submission will be sent to you soon.